# Hepatitis B Vaccine: Four Decades on

**DOI:** 10.3390/vaccines12040439

**Published:** 2024-04-18

**Authors:** Maria Mironova, Marc G. Ghany

**Affiliations:** Clinical Hepatology Research Section, Liver Diseases Branch, National Institute of Diabetes and Digestive and Kidney Diseases, National Institutes of Health, Bethesda, MD 20892-1800, USA; maria.mironova@nih.gov

**Keywords:** hepatitis B virus, hepatitis D virus, hepatocellular carcinoma, liver transplant, vaccination guidelines, birth dose vaccination

## Abstract

Hepatitis B virus is a substantial contributor to cirrhosis and hepatocellular carcinoma (HCC) globally. Vaccination is the most effective method for prevention of hepatitis B and its associated morbidity and mortality, and the only method to prevent infection with hepatitis D virus. The hepatitis B vaccine has been used worldwide for more than four decades; it is available in a single- or triple-antigen form and in combination with vaccines against other infections. Introduction of the vaccine and administration at birth led to sustained decline in mother-to-child transmission, chronic hepatitis B, and HCC, however, global birth dose coverage remains suboptimal. In this review we will discuss different hepatitis B vaccine formulations and schedules, vaccination guidelines, durability of the response, and vaccine escape mutants, as well as the clinical and economic benefits of vaccination.

## 1. Introduction

Hepatitis B virus (HBV) infection is a major cause of chronic liver disease and the leading cause of hepatocellular carcinoma (HCC) worldwide. Globally, the World Health Organization (WHO) estimates there are 296 million persons with chronic HBV infection and 820,000 HBV-related deaths annually [1].

Although effective treatment is available, none are curative. A safe and effective vaccine has been available since 1980, yet prevalence of chronic HBV infection has not declined substantially. Universal vaccination prevents both vertical and horizontal transmission of HBV, is associated with a reduction in morbidity and mortality from chronic hepatitis B, and has been shown to be cost-effective. Additionally, HBV vaccination can prevent infection with hepatitis D virus (HDV) and HCC, particularly childhood HCC. Indeed, the vaccine is considered one of the first anti-cancer vaccines. Consequently, the HBV vaccine is included in the national immunization schedules of 190 countries; however, coverage of the recommended birth dose (BD) vaccine remains suboptimal.

In this review we will discuss HBV vaccine development; formulations and schedules; guidance on who should receive vaccination, including special populations; durability of the response; vaccine escape mutants; and progress in HBV elimination since its introduction.

## 2. Hepatitis B Virus and Hepatitis B Disease

HBV is a partially double-stranded DNA virus, belonging to the *Hepadnaviridae* family. The complete virion (Dane particle) has a spherical double-shelled structure of ~42 nm in diameter. Two smaller spherical and filamentous particles can be observed by electron microscopy and consist of hepatitis B surface antigen (HBsAg) complexed with host lipids but lacking viral DNA. These particles are therefore non-infectious and vastly outnumber the infectious Dane particle. The viral genome has four overlapping open reading frames that encode for seven viral proteins: core, polymerase, e antigen, large, middle, and small surface antigen, and the x protein. Hepatocytes are the primary replication site of HBV. Replication occurs through an RNA intermediate, and the viral genome can integrate into the host genome. Based on sequence variability of the surface gene, eight well-established genotypes, A-H, have been identified, and two additional possible genotypes, I-J. There are four serotypes, *adw*, *ayw*, *ayr*, and *adr*, all of which share a common “a” determinant, which is targeted by neutralizing antibodies (anti-HBs). This explains the broad protection provided by HBV vaccine [2,3,4].

HBV is a blood-borne pathogen. Transmission can be vertical, also termed mother-to-child transmission (MTCT), or horizontal, through exposure to contaminated blood or body fluids between toddlers and young children, household contacts, during sexual intercourse, or via injection drug use, as well as nosocomial transmission [5]. In regions of high HBV prevalence there are geographical differences in the mode of HBV transmission. The predominant mode of transmission in Asia is vertical due to a high prevalence of hepatitis B e antigen (HBeAg) positivity among women of childbearing potential, whereas in sub-Saharan Africa the primary mode of transmission is horizontal between toddlers and children, with most being infected by school age. Horizontal transmission is the primary mode in low prevalence regions. The initial infection is typically asymptomatic and anicteric, particularly when infection is acquired vertically. Approximately 30% of adults may present with jaundice; acute liver failure is uncommon but occurs in both adults and children [6,7]. Development of chronic infection is age-dependent, occurring in >90% of persons less than one year of age, 30% in those less than five years of age, and less than 10% in adults. An estimated 25% of persons with chronic infection are at risk for developing cirrhosis [8]. Once cirrhosis develops, the risk of HCC is 2–4% annually. Notably, ~20% of HCCs develop in the absence of cirrhosis [9].

## 3. History of HBV Vaccine Development

The concept for developing a hepatitis B vaccine using HBsAg derived from human plasma was initially proposed by Blumberg and Millman in 1969 [10]. In the early 1970s, Krugman et al. tested the efficacy of a purified, heat-inactivated plasma containing HBsAg obtained from the blood of HBsAg-positive persons. It was about 70% protective, and two inoculations were shown to provide better protection compared to one [11]. Independently, Purcell and Gerin at the National Institute of Allergy and Infectious Diseases, Hilleman at Merck, and Maupas at the Institute de Virologie de Tours, isolated and purified HBsAg from the plasma of HBsAg-positive persons and tested immunogenicity and efficacy in chimpanzees [12,13,14]. Ultimately, the first plasma-derived vaccine, Heptavax-B, was manufactured by Merck and approved by the Food and Drug Administration in 1981 [15] (Figure 1).

The use of a plasma-derived vaccine posed several production challenges: the limited amount of suitable plasma available for vaccine production, purification costs, and potential contamination with other infectious agents. The advent of recombinant DNA technology led to the development of the first human recombinant vaccine in 1986, also by Hilleman through a collaboration between Merck and Chiron Corporation [16]. The HBsAg was expressed in vector-transformed *Saccharomyces cerevisae* yeast cells [17]. The early studies confirmed the safety and efficacy of the new recombinant vaccine; moreover, anti-HBs seroconversion rates were similar to the plasma-derived vaccine [18,19]. The first recombinant vaccine in Europe was Engerix-B, or HepB-Eng, produced by Glaxo Smith Kline (GSK).

The two recombinant vaccines, Recombivax-HB (Merck) and Engerix-B (GSK), have been used successfully for almost four decades. Safety, efficacy, and durability of protection have been demonstrated by numerous studies. However, achieving adequate and durable seroprotection in certain populations, such as older and immunocompromised individuals and patients with chronic kidney disease (CKD), remained a challenge. To improve rates of seroprotection in these populations and reduce the number of vaccine doses, two third-generation vaccines were developed. Heplisav-B, or Hep-CpG, containing the small S protein along with a novel cytosine phosphoguanine (CpG) immunostimulatory adjuvant (CpG 1018), obtained regulatory approval in the U.S. in 2017 [20]. PreHevbrio (initially licensed as Sci-B-Vac), containing the small S protein and enhanced by addition of large (pre-S1) and middle (pre-S2) proteins, was approved in 2021. A two-dose course of Heplisav-B was associated with a faster and more durable seroprotection compared to conventional recombinant vaccines, especially in populations with low response rates [20,21,22]. Studies with a three-dose course of PreHevbrio demonstrated that seroprotection rates were non-inferior to conventional recombinant vaccines in the general adult population. One study of patients with inflammatory bowel disease (IBD) reported higher seroprotection rates compared to that of mono-antigen vaccines [23,24].

## 4. Novel Vaccine Approaches

While current vaccines are effective, the requirement for two or three doses remains a barrier to delivery and more widespread use. Current research is aimed at developing vaccines that are more immunogenic, using novel adjuvants thereby increasing seroprotection rates, particularly among difficult-to-vaccinate populations. A marked antibody response has been observed with the ferritin nanoparticle-preS1 vaccine, which holds the potential for use both as a prophylactic and a therapeutic vaccine [25]. A plasmid DNA fusion vaccine encoding mouse DEC-205 single-chain fragment variable (mDEC-205-scFv) to direct antigens to dendritic cells (DCs), through the DC-specific surface molecule DEC-205, was linked with the HBsAg [26]. This vaccine was shown to induce a robust antiviral T cell and antibody immunity against HBsAg in HBV transgenic mice. Another approach is to use HBsAg virus-like particles (VLP) as biotemplates to synthesize silica-adjuvanted VLP@Silica nanovaccines, which can induce both a humoral and cellular immune response [27]. The success of COVID-19 vaccines has spurred further research into mRNA vaccines for hepatitis B. While initially pursued as a promising therapeutic vaccine, mRNA vaccines can induce more robust immune response and may be used for prophylaxis [28]. Strategies are also focused on utilizing novel delivery systems, including adenoviral and yeast vectors, to improve vaccine effectiveness. Adenoviral vectors have been noted for eliciting a strong antibody response, which is particularly promising for individuals with low seroconversion rates. Moreover, adenovirus vector-based vaccines might offer advantages in terms of manufacturing ease and reduced storage costs compared to recombinant vaccines, potentially enhancing their adoption in low- and middle-income countries [29]. Another novel approach to antigen delivery is to coat soluble microneedle arrays with mannose-modified poly lactic-co-glycolic acid nanoparticles. Proof of principle for this approach was demonstrated in a mouse model [30].

Alternate routes of administration are being explored, including intranasal and intradermal routes. An intranasal vaccine containing both HBsAg and hepatitis B core antigen (HBcAg) mixed with carboxyl vinyl polymer (CVP-NASVAC) as a viscosity enhancer, was shown to induce anti-HBs and anti-HBc production, and led to a substantial increase in HBs- and HBc-specific cytotoxic T-lymphocyte responses after CVP-NASVAC administration in prior hepatitis B recombinant vaccine non-responders. Another approach is a microneedle patch, a promising vaccine delivery method that does not require a cold chain system and is convenient for use in areas where healthcare resources are scarce; it has been shown to elicit a stronger immune response than traditional intramuscular administration [31,32].

## 5. Indications for Use-Target Populations

Taiwan, as an endemic country with an HBsAg prevalence of 15–20%, prior to introduction of the hepatitis B vaccination, was the first country to introduce an HBV vaccination program in 1984. Vaccination was initially offered to infants born to HBsAg-positive mothers but was later expanded to all infants in 1986 [33]. Among non-endemic countries with low HBsAg prevalence, vaccination was recommended for infants, children, and adults at increased risk for HBV. These included men who have sex with men (MSM), household contacts of infected individuals, persons requiring hemodialysis (HD), those with thalassemia, healthcare personnel, and institutionalized persons. Identification of high-risk individuals was a challenge due to the stigma associated with certain risks. Moreover, infection rates among heterosexual individuals, people who inject drugs (PWID), females, and adolescents were steadily increasing [34]. Failure to decrease the incidence of disease highlighted the need for a new strategy. In 1991, the World Health Organization (WHO), recognizing the public health burden of chronic hepatitis B, recommended the integration of HBV vaccination into National Immunization programs globally by 1997 [35]. Furthermore, in 1994, they set a target of reducing new infections among children by 80% by 2001. In 2009, the WHO recommended timely administration of BD HBV vaccine (within the first 24 h of life) to all newborns [36].

In 1991, the United States recommended universal HBV vaccination of all infants and young children. Recommendations for vaccination were further expanded to include catch-up immunization of previously unvaccinated children and adolescents, as well as vaccination of unvaccinated adults at increased risk for infection, as part of a national strategy to eliminate transmission of HBV in the U.S. [37]. In 2023, the Centers for Disease Control and Prevention (CDC) updated their recommendations, advising hepatitis B vaccine to all adults through age 59 years who have not been vaccinated or whose vaccination status is unknown, and adults age 60 years or older with risk factors for hepatitis B infection or who desired protection (Table 1). Additionally, the CDC recommended that all adults should be screened for HBV at least once in their lifetime [38,39].

## 6. Immunization Schedules

The HBV vaccine is available as a single-antigen formulation which may be administered at birth, infancy, childhood, and adulthood, and as a multi-antigen vaccine only for adults. It is also available as part of combination vaccines for infant and childhood immunizations, including DTaP-HepB-IPV (Pediarix), consisting of diphtheria, tetanus toxoid, and acellular pertussis adsorbed (DTaP); inactivated polio vaccine (IPV) and hepatitis B (recombinant); DTaP-IPV-Hib-HepB (Vaxelis), a hexavalent preparation containing diphtheria, tetanus toxoid, and acellular pertussis adsorbed; inactivated polio vaccine, haemophilus influenzae type b (Hib), and hepatitis B (recombinant); and in combination with hepatitis A vaccine for both children and adults (HepA-HepB, Twinrix) [40]. The HBV vaccine is not known to affect the immunogenicity of other vaccines [41,42].

The usual adult dose of recombinant hepatitis B vaccine is 10–40 μg of HBsAg, and the pediatric dose is 5–10 μg of HBsAg. Immunocompromised patients may be vaccinated with higher doses. The vaccine is administered by intramuscular injection into the anterolateral aspect of the thigh in infants or the deltoid muscle in older children and adults [40]. Intradermal injection may be a potential alternative route of administration. It requires a smaller amount of HBsAg per injection, and therefore is cost-saving. It has been shown to induce higher rates of seroconversion in populations with low response, such as patients with IBD and those on HD [43,44].

For primary prophylaxis, the WHO recommends two vaccination schedules depending on individual national immunization programs. The standard schedule for infants is three doses at 0 (BD vaccine), 1, and 6 months. A four-dose vaccination schedule includes a BD vaccine followed by three doses of a combination vaccine as per the combination vaccine schedule. (Table 2 and Table 3). In Africa, only 14 out of 47 countries have introduced BD vaccination into their immunization calendars [45], with a majority of countries starting immunization at 6 weeks. Although vertical transmission is a less common mode of transmission in Africa, delay in vaccine administration may contribute to incident infection and timely administration of BD vaccination is preferred.

A delayed schedule for children and adults is similar to the three-dose schedule for infants. If the second or third dose is delayed it should be administered as soon as possible, but the vaccination series should not be interrupted [40]. Increasing the interval between the first two doses has little effect on the immunogenicity or final level of anti-HBs. The interval between the second and third doses should be a minimum of 4 weeks. Longer intervals between the last two doses result in higher final anti-HBs levels but might increase the risk of acquisition of HBV infection among persons who have a delayed response to vaccination. A study from Korea assessed whether delay between the BD and the second dose was associated with higher perinatal transmission in infants born to HBsAg-positive mothers. There was no difference between those who received a second dose within an interval of 4–8 weeks and after ≥8 weeks [47]. The third dose of recombinant vaccine confers the maximum level of seroprotection, acts primarily as a booster, and appears to provide optimal long-term protection [48].

For post-exposure prophylaxis, HBV vaccine schedules depend on the type of setting (occupational versus nonoccupational), immunization status of the exposed person, and HBsAg status of the contact (Table 4).

An accelerated schedule can be considered in certain groups with a need for rapid immunity. The benefit of an accelerated schedule has been demonstrated in immunocompromised persons, patients with cirrhosis [49], patients on HD [50], PWID [51], and high-risk pregnant women [52]. Moreover, accelerated immunization may have a better compliance.

## 7. Efficacy and Seroprotection Rates

The marker of immunity against HBV is presence of immunoglobulin G (IgG) to HBsAg (anti-HBs) after vaccination. Among vaccinated persons, anti-HBs is the only detectable antibody, whereas those previously exposed to HBV will also have detectable anti-HBc (antibody to core antigen and may have antibody to HBeAg (anti-HBe)). An anti-HBs level of >10 mIU/mL is considered protective against infection. After a full vaccination series with recombinant vaccines, seroprotection rates in children reach 98–100%, and 90–95% in adults [53,54]. Up to 5–10% of immunocompetent adults do not mount protective levels of antibody after the initial vaccination series. Among factors associated with decreased response are obesity [55], smoking [56], and age older than 40–50 years [57]. Seroprotection rates in adults are higher with the newer vaccines compared to Engerix; PreHevbrio vs. Engerix (91.4 versus 76.5%, respectively), and Heplisav-B vs. Engerix (95.4% versus 81.3%, respectively). This is particularly evident in adults older than 50 years, making them an attractive choice for more difficult-to-vaccinate persons [58,59].

## 8. Durability of Seroprotection and Need for Booster Doses

Long-term follow-up studies of persons from endemic regions who were vaccinated as infants have reported loss of anti-HBs in 50–70% of persons, 15–30 years later. Long-term follow up data from the Taiwanese immunization program reported that seroprevalence of anti-HBs among persons born after the start of nationwide immunization was 48.4% after 20 years, and 44% after 30 years [60,61]. Bruce et al. studied persistence of anti-HBs in an Alaskan Native population 35 years after initial vaccination. Fifty-one percent had anti-HBs levels ≥ 10 mIU/mL at 30 years, and forty-seven percent had anti-HBs ≥ 10 mIU/mL at 35 years. Persons vaccinated at ages 5–19 years were more likely to have protective antibodies 30 years later compared to persons vaccinated at other ages. Most subjects (88%) with anti-HBs levels < 10 IU/mL responded to a booster dose developing anti-HBs levels > 10 IU/mL [62,63]. Long-term follow-up studies of persons vaccinated as adults report higher detection of anti-HBs. Van Damme et al. reported that 90% of persons vaccinated as adults had anti-HBs levels ≥ 10 mIU/mL 30 years after vaccination. Following challenge with a booster dose, all subjects mounted an anamnestic response with a marked increase in HBsAg-specific memory B and CD4+ T cells [64]. Another study of healthcare workers vaccinated as adults reported that 77% had anti-HBs levels ≥ 12 IU/mL a mean of 18 years later. Among those with an inadequate anti-HBs level, 94% developed protective levels of anti-HBs by 3 weeks following a booster dose [65]. A review of 83 associations pooled from multiple meta-analyses showed that long-term vaccine efficacy is enhanced with a booster dose and use of HepB-CpG as the primary vaccine, although duration of follow-up is shorter [66].

A meta-analysis of studies conducted to assess seroprotection rates, based on the timing of vaccination in children, reported that the pooled seroprotection rate in children under 5 years old in Africa with four doses (i.e., a BD and three doses) was significantly higher than three doses, highlighting the importance of BD [67]. Another study from Pakistan compared seroprotection rates 8 weeks after completion of a vaccination series in infants who received BD and three doses of combination vaccine to those who received combination vaccines only. There was a dramatic difference observed between the BD and non-BD groups, 95.9% vs. 58.8%, respectively [68]. In a study from the Philippines, seroprotection rates one month after vaccination series with DTaP-HBV/Hib was higher in infants who also received a BD vaccine, 94.3% compared to 87.6% in infants who received the combination vaccine only [69]. Taken together, these data provide support for the importance of incorporating of BD into national immunization calendars worldwide.

Currently, a booster dose, or subsequent vaccination after a primary vaccination series, is not recommended by the WHO or CDC for fully vaccinated immunocompetent individuals. Despite waning anti-HBs levels, the rapid and robust response to a booster dose among the majority (~90%) of healthy adults with anti-HBs < 10–12 mIU/mL suggests persistence of an amnestic response up to 30 years following primary vaccination. This response confers long-term protection against infection. How long immune memory will last is unknown and will require additional long-term follow-up studies. Booster doses, however, are recommended for persons receiving HD or who are otherwise immunocompromised if anti-HBs levels decline <10–12 mIU/mL because immunologic memory is only reliable in healthy persons.

There have been increasing calls to recommend a booster dose, especially in young individuals, based on several studies on blood donors in China suggesting that young individuals who have been vaccinated remain susceptible to occult HBV infections, as indicated by presence of isolated anti-HBc, anti-HBc, and anti-HBs, and even anti-HBs alone, together with intermittent low-level detection of HBV DNA suggesting vaccine failure in some cases. This has been attributed to waning immunity with low or undetectable anti-HBs, failure to complete the vaccination schedule, or lack of evidence of vaccination leading some to recommend a booster vaccine at adolescence (before the start of sexual contacts). The adoption of such a strategy remains controversial and is not currently recommended [70,71,72]. Moreover, a study from Thailand compared long-term seroprotection during a 20-year follow-up after vaccination in infancy, where half of the subjects received a booster. The vaccine provided long-term seroprotection regardless of whether a booster dose was administered, although there was evidence of HBV exposure in the cohort. Breakthrough infections in the first decade usually occurred due to household exposure, while those in the second decade occurred with the beginning of adolescent behaviors [73].

Persons with isolated anti-HBc are at risk for HBV reactivation if receiving immunosuppressive therapy. The American Association for the Study of Liver Diseases hepatitis B guidance does not currently recommend vaccination to boost anti-HBs levels to >100 IU/mL as a strategy to prevent HBV reactivation among persons scheduled to receive immunosuppressive therapy. The European Association for the Study of Liver Diseases hepatitis B guidelines do recommend that patients without serological evidence on hepatitis B, and who are candidates for immunosuppression, should be vaccinated [74].

## 9. Safety and Side Effects

Both the mono- and triple-antigen hepatitis B vaccines, and the mono-antigen vaccine in combination with other vaccines, have excellent safety profiles. Several case reports published in the mid 1990s raised concerns for a possible link between hepatitis B vaccination and the exacerbation of multiple sclerosis and other central nervous system demyelinating disorders. These reports led to a temporary suspension of the school vaccination program in France [75,76]. Subsequent studies confirmed that there was no increased risk of multiple sclerosis relapse or demyelination after hepatitis B vaccination [75,77,78,79]. Adjuvants added to hepatitis B vaccines were also claimed to be causally associated with autoimmune disorders, such as rheumatoid arthritis, systemic lupus erythematosus, and thyroid disease, however, there was no evidence to support this [80].

The hepatitis B vaccine is safe for administration during pregnancy [81,82]. Vaccination during pregnancy is recommended for women with no evidence of immunity and at high risk for infection [83]. Hepatitis B vaccines containing the CpG adjuvant are currently not recommended in pregnancy due to lack of available safety data. Preliminary data suggest a similar safety profile and pregnancy outcomes in women who became pregnant after vaccination with HepB-CpG and after HepB-alum formulations [84].

The most common side effects include injection site pain, swelling, erythema, and mild systemic symptoms such as myalgia or low-grade fever [85,86]. Vaccine tolerability in neonates may be improved if administered during breastfeeding or skin-to-skin contact [87,88].

## 10. Management of Non-Responders to an Initial Vaccine Series

Shortly after introduction of the hepatitis B vaccine, it was recognized that several populations had a decreased immunological response. Ironically, these were mostly persons at high risk of HBV infection: patients receiving HD and immunocompromised individuals, either by virtue of underlying disease or due to immunosuppressive therapies. Among PWID and MSM, achieving protective antibody levels was challenging due to frequent occurrence of HIV infection. Notably, half of HIV-positive MSM failed to develop anti-HBs [89]. People with hepatitis C infection similarly demonstrated impaired immune responses [90]. Effectiveness of the HBV vaccine in people with chronic liver disease, particularly cirrhosis, was also suboptimal, with an overall response rate of 47% [91].

To ensure adequate protection for at-risk individuals, post-vaccination testing may be performed 1–2 months after the final dose. If anti-HBs is not detected or the level is <10 IU/L, revaccination with a booster dose or a complete vaccination series should be performed. Currently, annual testing is recommended for HD patients and may be considered in individuals with ongoing risk for HBV infection [59]. Specific management of groups with decreased response to hepatitis B vaccine is discussed below.

### 10.1. Chronic Kidney Disease and Hemodialysis

The estimated prevalence of HBV infection among people receiving HD is 7.32%. The rate is higher in developing compared to developed countries [92]. The duration of receiving HD is an important risk factor for HBV infection [93]. Prevention of HBV in this population is challenging. Chronic kidney disease (CKD) patients have impaired immunological response to vaccination, which may be explained by reduction in some subsets of T-cells, and faster anti-HBs decay compared to healthy individuals [94]. Several strategies have been shown to be effective in improving immune response in CKD; these include use of adjuvanted vaccines and higher doses of antigen: 40 or 60 μg instead of 20 μg [95,96,97]. The presence of several modifiable factors, such as diabetes mellitus, inadequate nutrition, low hemoglobin, and parathyroid hormone, are associated with decreased response to hepatitis B vaccine in CKD. Correction of these factors may potentially improve seroprotection rates [98]. HepB-CpG has been shown to be safe, well tolerated, and it induces a high rate of seroprotection among adults receiving HD [99]. After four doses of HepB-CpG at week 20, the proportion of participants with anti-HBs levels ≥ 10 mIU/mL was 89.3% and the proportion with anti-HBs ≥ 100 mIU/mL was 81.3%. In a follow-up study, the durability of seroprotection in participants with CKD was similar in those who received HepB-CpG compared to those who received HepB-Eng. However, among participants who achieved anti-HBs levels > 100 mIU/mL, levels ≥ 100 mIU/mL persisted for longer in participants who received HepB-CpG compared to HepB-Eng [100].

### 10.2. Human Immunodeficiency Virus

HIV and HBV share similar routes of transmission, which explains the high prevalence of HBV infection among people living with HIV (PLWH). The risk of developing chronic HBV is six-fold higher in PLWH compared to those without HIV [101]. This is because resolution of HBV infection is dependent on a robust HBV-specific T-cell response, which may be lacking in immunocompromised persons. In a meta-analysis of 17 studies including 1,821 PLWH, the overall pooled response to recombinant hepatitis B vaccine was 71.5%. Doubling the antigen dose (from 20 μg to 40 μg) compared to standard dose vaccine (75.2% versus 65.5%) and a four-dose schedule versus three-dose schedule (89.7% versus 63.3%) were associated with an improved vaccination response. Additionally, people with CD4+ T-cells > 500 cells/mm^3^ had better immune response [102]. Therefore, one strategy to increase seroprotection rates among HIV-positive persons failing a standard hepatitis B vaccine series is to re-vaccinate using a higher dose regimen (40 μg of HBsAg) [102,103]. The other more promising approach is to use HepB-CpG for the primary vaccine series. Among 68 HIV-positive individuals, 100% achieved seroprotection 4 weeks after the vaccine series; of these, 88% had anti-HBs levels > 1000 mIU/mL [104]. In another retrospective analysis, HepB-CpG was associated with a significantly higher rate of seroprotection compared with HepB-alum-based hepatitis B vaccines, 93.4% compared with 57.6%, respectively [105].

### 10.3. Inflammatory Bowel Disease

The prevalence of HBV in IBD patients is similar to the general population, however, the vaccine seroprotection rate is lower, ranging between 12% and 42%, which can be partially explained by disease-related impaired immunity [106,107]. Immunosuppressive and biologic therapies (steroids, immunomodulators, including thiopurines and methotrexate, and monoclonal antibodies to tumor necrosis factor (TNF)-alpha) [107,108] have transformed the management of IBD but, at the same time, their use is associated with a lower probability of response to recombinant hepatitis B vaccine [107]. In a meta-analysis of 21 studies including 2602 patients, the pooled rate of seroprotection (anti-HBs > 10 mIU/mL) was 62%. Gender, IBD subtype, and disease activity did not affect the response rate. However, use of immunosuppression and anti-TNF agents were predictors of poor immune response compared to no immunosuppressive therapy. Durability of response is also decreased in IBD. In one study, 20% of initially immune patients had negative anti-HBs levels within 12 months. Antibody loss was increased in those with anti-HBs levels < 100 IU/L after the vaccination [107]. Therefore, it is recommended to vaccinate against hepatitis B before starting the immunosuppressive agents.

### 10.4. Preterm Infants

Preterm birth and low birth weight (LBW) < 2000 g have been associated with a decreased response to hepatitis B vaccine [109]. Both the WHO and CDC recommend that LBW infants should receive the BD vaccine, but it should not count towards the vaccine series. BD vaccine should be followed by the three-dose vaccine series beginning at the age of 1 month. The final dose of the series should not be administered before age 24 weeks. Vaccination of LBW infants born to HBsAg-negative mothers can be delayed until discharge from the hospital or 1 month of age [46].

## 11. Hepatitis B Vaccine Escape Mutants

HBV circulates as a viral quasispecies due to the lack of proofreading capacity of the HBV reverse transcriptase. Therefore, immune pressure following vaccination may select variants that can escape neutralizing immunity afforded by vaccination and represent a potential threat for the global effectiveness of hepatitis B vaccination. In support of this concern, vaccine-induced escape mutations have been reported in persons who have become infected despite vaccination, [110,111,112] (Table 5). Several studies have examined whether the prevalence of vaccine escape mutants is increasing in the population with chronic hepatitis B. One study from Taiwan examined the prevalence of “a” determinant mutants (HBsAg amino acids 110–160) (vaccine mutants) among HBV-infected children before (1984), 5 years after (1989), and 10 years after (1994) universal vaccination was introduced. The prevalence of “a” determinant mutants increased from 8 of 103 (7.8%) in 1984 to 10 of 51 (19.6%) in 1989 and to 9 of 32 (28.1%) in 1994 and was higher in those fully vaccinated than unvaccinated [113]. However, a study from China conducted in the post-vaccination era reported that the prevalence vaccine escape mutants remained relatively stable from 2005 to 2013, with an annual prevalence that ranged between 6.1% and 12.8% [114]. Another study from Italy detected hepatitis B vaccine escape mutants in 17.7% of patients with genotype D. Of more concern was a rising prevalence of complex mutations (defined as ≥2 vaccine escape mutations) over time from 0.4% in 2005–2009 to 3.0% in 2010–2014 and 5.1% in 2015–2019 [115]. Despite these findings, there have not been widespread infections occurring in vaccinated individuals. While vaccine escape mutations may account for some vaccine failures, the prevalence of vaccine escape mutations is lower than the rate of vaccine failures, suggesting that these mutations cannot account for all immunoprophylactic failures in infants [116]. Long-term monitoring for vaccine escape mutants should continue to ensure that these variants do not pose a risk to HBV elimination programs.

## 12. Clinical and Economic Impact of Hepatitis B Vaccination

Vaccination is the cornerstone of HBV elimination efforts through prevention of transmission. Therefore, in addition to universal vaccination of infants beginning at birth, the Advisory Committee on Immunization Practices (ACIP) recommends “catch-up” vaccination among all adults aged 19–59 years and adults > 60 years with risk factors or without risk factors for hepatitis B who seek protection. Substantial progress has been made toward achieving these goals. In 2022, the number of countries that incorporated the HBV vaccine into their national vaccination calendars was 190, and 113 countries introduced the recommended BD vaccine [117]. The hepatitis B vaccine has been shown to be effective in reducing rates of acute and chronic hepatitis B, HBV-related cirrhosis, HCC, and deaths, as well as the number of carriers (Table 6).

Remarkable evidence of vaccine effectiveness comes from the Qidong Hepatitis B Intervention Study conducted in Qidong, China, between 1983 and 1990. Approximately 41,000 newborns were assigned to the vaccination arm, while approximately 41,000 were in control group. After 37 years of follow-up, the hepatitis B vaccine was shown to have 72% efficacy against the development of liver cancer, 70% efficacy against death from liver cancer, and a 64% reduction in liver-related death [127]. The incidence rate of HCC among vaccinated persons was 0.46/100,000 compared to 1.61/100,000 in an unvaccinated control group, and death from liver disease was 0.83/100,000 in the vaccinated arm compared to 2.23/100,000 in the unvaccinated arm. Qu et al. published additional insights on incidence of infant fulminant hepatitis, which was 10.5/100,000 in the vaccinated group compared to 27.5/100,000 in controls [128].

Similar results were seen following the introduction of the hepatitis B immunization program in Taiwan, a hyperendemic area with a HBsAg prevalence of 9.8% in the pre-vaccine era. The current vaccination coverage rate has reached 98% [33]. As evidence of the success of the vaccination program, the seroprevalence of HBsAg among those born before and after mass vaccination was 7.7% and 0.4%, respectively [129]. A direct effect of this reduction in seroprevalence was a marked reduction in complications of chronic hepatitis B. An early landmark study conducted in 1997 by Chang et al. showed a dramatic reduction in incidence of childhood HCC from 0.70 per 100,000 (1981–1986) to 0.33 per 100,000 (1990–1994) [130]. The most recent data demonstrated that rates of mortality from infant fulminant hepatitis have decreased by more than 90%, and from HCC and liver disease by more than 90% [120]. Furthermore, the incidence rate of HCC has decreased by more than 80% in the population aged 5–29 years and likely would be effective in older persons as long as there is evidence of seroprotection [131]. The vaccine may have therapeutic potential in reducing HBsAg levels and inducing HBsAg loss in persons with low HBsAg levels [132].

### 12.1. Reduction in Hepatitis D Virus Infections

An added value of hepatitis B vaccination is protection against HDV, which requires co-infection with HBV to provide its envelope proteins for viral entry [133]. Although the data are limited, there is a trend toward reduction of HDV prevalence associated with implementation of HBV vaccine programs. A study by Chen et al. analyzed HDV prevalence before and after 1997, the year when the HBV vaccine was recommended for inclusion in the Expanded Program of Immunization in all countries by the WHO. Global prevalence of HDV was higher in 1977–1996 than in 1997–2016, however, an opposite trend was observed in some countries, like Japan and Gabon [134]. A study from Turkey demonstrated an overall decrease in HDV prevalence from 8.3% observed in years before 1999 to 5.5% after 2010 [122]. In Italy, the decline in incidence of acute hepatitis D occurred in parallel with the decrease in incidence of acute hepatitis B, suggesting an effect of hepatitis B vaccination on both infections [123].

### 12.2. Cost-Effectiveness of HBV Vaccination

Data on the clinical and economic outcomes of hepatitis B vaccination are important for supporting global and regional funding and efforts for HBV elimination. Many studies conducted in low- and middle-income countries have determined that administration of hepatitis B vaccine is cost-effective or cost-saving. Data from China, which accounts for 30% of the global HBV burden, estimated that in 2020, vaccination of 10 million infants born in China (where newborns have received vaccines at no cost since 2002) will cost 0.52 billion US dollars, however, it will save approximately 120 billion dollars in future healthcare costs and lost productivity [135].

HBV prevention strategies targeting perinatal transmission are considered very cost-effective, with an incremental cost-effectiveness ratio of $6957 per quality-of-life-adjusted life year saved when comparing hepatitis B vaccine and hepatitis B immunoglobulin (HBIG) for infants born to HBsAg-positive mothers and universal infant vaccination prior to hospital discharge to a strategy of universal infant vaccination prior to hospital discharge alone [136]. The cost benefits of universal adult HBV vaccination in the US were estimated using simulated modeling. Coverage of 50% of the adult population will reduce the number of acute infections by almost 25% and avert a loss of nearly $227,000 per each acute infection [137].

## 13. Challenges in Hepatitis B Elimination and Future Steps

MTCT is the most common route of transmission worldwide. Thus, disrupting MTCT is crucial for the elimination of HBV. Modeling studies have reported that prenatal care for HBsAg-positive mothers and their children remains suboptimal in 2022. Only 3% of pregnant women with high viral load (>200,000 IU/mL) receive antiviral treatment during pregnancy, only 46% infants receive timely BD vaccination (Figure 2), and 14% of infants receive HBIG [138]. In addition to reducing MTCT, greater efforts are needed to increase BD and “catch-up” vaccination, particularly in high prevalence areas. The region with the lowest BD coverage is sub-Saharan Africa, where the estimated HBsAg prevalence ranges from 6.3–10% [138,139]. Barriers to providing BD vaccination occur at all levels: policy, facility and logistics, and community, (Table 7). Logistical constraints are mostly associated with births occurring in remote areas and outside health facilities [139]. Strategies to overcome these challenges include improving healthcare infrastructure, promotion of awareness campaigns, and policies that facilitate access to vaccination, such as maintaining a consistent and reliable vaccine supply, proper vaccine storage under controlled-temperature chain protocol when cold chain use is impossible, and use of prefilled auto-disposable devices to facilitate vaccine administration [140]. In tandem with these solutions, there is a critical need for more robust systems for collecting and analyzing vaccination and infection rate data in order to help identify areas for improving vaccination strategies and targeting interventions more effectively.

The focus of the WHO Global Health Sector Strategies 2022–2030 is elimination of HBV infection through universal access to BD vaccine, improved care for pregnant women to prevent vertical transmission, and investment in primary prevention. The key goals are a decrease in the number of new infections from 20 to 2 per 100,000 a year, reduction in mortality due to HBV to 310,000 per year, and timely administration of BD and other preventive measures to 90% of children born to HBV-infected mothers [142].

## 14. Conclusions

The development of a safe and effective hepatitis B vaccine has provided the means to prevent and eliminate HBV. Significant progress has been made towards elimination of HBV since the inclusion of the hepatitis B vaccine in national universal immunization programs. The prevalence of HBV infection among those <18 years of age has declined dramatically. However, the burden of HBV-related chronic liver disease, cirrhosis, and HCC remains high. Addressing inequalities in universal access to birth dose vaccine for all infants, screening, perinatal care for HBsAg-positive pregnant women, and catch-up vaccination for adults are important steps toward achieving HBV elimination.

## Figures and Tables

**Figure 1 vaccines-12-00439-f001:**
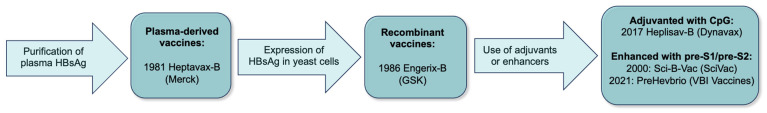
Timeline of hepatitis B vaccine development (HBsAg, hepatitis B surface antigen; CpG, cytosine phosphoguanine).

**Figure 2 vaccines-12-00439-f002:**
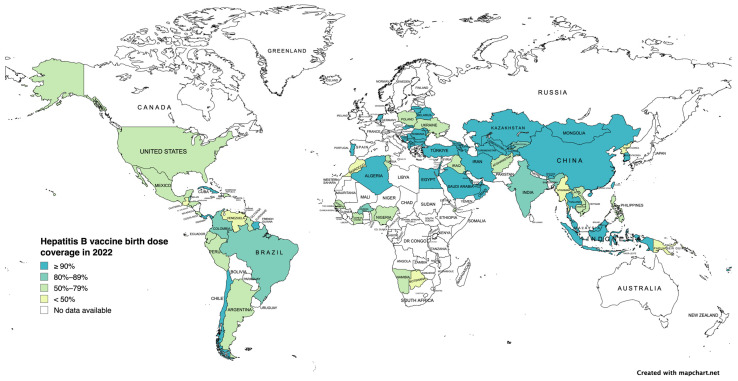
Hepatitis B vaccine birth dose coverage in 2022 (adapted from the World Health Organization Hepatitis B Vaccination Coverage Dashboard) [141].

**Table 1 vaccines-12-00439-t001:** Persons who should receive hepatitis B vaccine in the United States.

Populations for Whom Hepatitis B Vaccine is Indicated
The following persons should receive hepatitis B vaccination:
All infants
Children and adolescents not previously immunized
Adults 19–59 years old not previously immunized
Adults older than 60 years with risk factors for infection
The following groups may receive hepatitis B vaccination:
Adults older than 60 years without risk factors for infection
Persons at risk for infection by sexual exposure:
Sex partners of persons who are positive for HBsAg
Men who have sex with men
Persons seeking evaluation or treatment for sexually transmitted infection
Sexually active persons who are not in a long-term, mutually monogamous relationship
Persons at risk for infection by percutaneous or mucosal exposure to blood:
Persons who inject drugs
Household contacts of persons who are positive for HBsAg
Healthcare and public safety personnel
Persons receiving dialysis
Staff and residents of facilities for persons with developmental disabilities
Persons with diabetes at the discretion of the treating clinician
Others:
Persons with hepatitis C infection or HIV
Persons with chronic liver diseases or diabetes
Travelers to countries with HBsAg prevalence ≥2%
Persons who are incarcerated
Adults older than 60 years without known risk factors for hepatitis B infection may receive HBV vaccines

HBV, hepatitis B virus; HBsAg, hepatitis B surface antigen. Adapted from Recommendations of the Advisory Committee on Immunization Practices, United States, 2022 [39].

**Table 2 vaccines-12-00439-t002:** WHO recommendations for HBV vaccination of infants.

	Dose and Timing			
Schedule	Birth Dose	1st Dose	2nd Dose	3rd Dose
Three-dose	Within 24 h	Not counted	A minimum of 4 weeks after the birth dose	A minimum of 4 weeks after the 2nd dose
Four-dose	Within 24 h	According to combination vaccine schedule	According to combination vaccine schedule	According to combination vaccine schedule

Adapted from the World Health Organization position paper on hepatitis B vaccines, 2017 [40].

**Table 3 vaccines-12-00439-t003:** Most common immunization schedules for primary prevention of hepatitis B in the United States. Recommendations provided for infants ≥ 2000 g born to HBsAg-negative mothers.

Population	Vaccine	1st Dose	2nd Dose	3rd Dose	4th Dose
Infants	Monovalent, Engerix-B, or Recombivax-HB	Birth dose within 24 h	1 month	6 months	
	Monovalent, Engerix-B, or Recombivax-HB + monovalent/combination Pediarix: HBV + DTaP + IPV or Vaxelis: HBV + DTaP + IPV + Hib	Birth dose within 24 h Monovalent	2 months Monovalent/combination	4 months Monovalent/combination	6 months Monovalent/combination
Unvaccinated children 1–18 years old	Monovalent Engerix-B or Recombivax-HB	Now/0 months	1 month	6 months	
Adults 19–59 years old and older than 60 years at risk for HBV infection	Monovalent Engerix-B or Recombivax-HB	Now/0 months	1 month	6 months	
	Monovalent Heplisav-B	Now/0 months	1 month		
	Trivalent PreHevbrio	Now/0 months	1 month	6 months	
	Monovalent/combination Twinrix: HAV + HBV	Now/0 months	1 month	6 months	

DTaP, diphtheria-tetanus-acellular pertussis; IPV, inactivated polio vaccine; HAV, hepatitis A virus; HBV, hepatitis B virus; Hib, *Hemophilus influenza* type b. Adapted from Recommendations of the Advisory Committee on Immunization Practices, 2018 [46] and 2022 [39].

**Table 4 vaccines-12-00439-t004:** Hepatitis B vaccine used for post-exposure prophylaxis.

Setting	Vaccination Status of the Exposed Person	HBsAg Status of a Source	
Occupational		Positive/Unknown	Negative
	Documented complete series + anti-HBs ≥ 10 mIU/mL	No treatment.	No treatment.
	Documented two complete series + anti-HBs < 10 mIU/mL (nonresponse)	First dose of HBIG within 24 h, followed by the second dose of HBIG in 1 month.	No treatment.
	Documented complete series + anti-HBs < 10 mIU/mL	HBIG simultaneously with first dose of HBV vaccine (within 24 h), followed by completion of revaccination series.	Single dose of HBV vaccine. Anti-HBs testing should be repeated in 1–2 months. Revaccination series should be completed if anti-HBs remains <10 mIU/mL.
	Unvaccinated or incomplete series	HBIG simultaneously with the first dose of HBV vaccine (within 24 h), followed by completion of vaccination series.	Vaccination series per immunization schedule.
Nonoccupational		Positive	Unknown
	Documented complete vaccination series	Single dose of HBV vaccine.	No treatment.
	In process of being vaccinated	HBIG + completion of vaccine series as scheduled.	Completion of vaccine series as scheduled.
	Unvaccinated	HBIG simultaneously with the first dose of HBV vaccine (within 24 h), followed by completion of vaccination series.	First dose of HBV vaccine within 24 h, followed by completion of vaccination series.

HBIG, hepatitis B immunoglobulin; HBsAg, hepatitis B surface antigen; HBV, hepatitis B virus. Adapted from Recommendations of the Advisory Committee on Immunization Practices, 2018 [46].

**Table 5 vaccines-12-00439-t005:** List of most common vaccine escape mutant within the “a” determinant region of the S-gene.

Amino Acid Position	Wild Type	Mutant
114	T	R
118	T	A
120	P	T
123	T	N
126	I/T	N/A
129	Q	H
130	G	R
133	M	L
141	K	E
142	P	S
143	T	M
144	D	H/A
145 *	G	R

* Most common vaccine escape mutation.

**Table 6 vaccines-12-00439-t006:** Global impact of hepatitis B vaccination.

Geographic Region	Country	Years	Metric	Impact
Africa	Burkina Faso	From 1996–2001 to 2012–2017	Prevalence of HBV infection	Decrease from 12.80% to 11.11% [118]
Asia	Iran	1993–2004	Prevalence of HBV infection	Decrease from 2.5% to 1.7% [119]
	Taiwan	1977–2011	Mortality from infant fulminant hepatitis among individuals 5–29 years old	Decrease from 5.76/100,000 to 0.03/100,000 persons/years
			Mortality from HCC	Decrease from 0.70/100,000 to 0.08/100,000 persons/years
			Incidence of HCC	Decrease from 1.14/100,000 to 0.09/100,000 persons/years [120]
	Republic of Korea	1999–2016	Incidence of pediatric HCC	Decreased from 0.12/100,000 to 0/100,000
		1980s-2016	HBsAg seroprevalence among teenagers	Decrease from 7.0% to 0.3% [121]
Europe	Turkey	1990–2017	Incidence of acute HBV in children < 5 years old	Decrease from 6.2/100,000 to 0.1/100,000
		1991–2004	Prevalence of HBV infection	Decrease from 5.2% to 2.1% [122]
	Italy	1987–2019	Incidence of acute hepatitis B infection	Decrease from 10/100,000 to 0.39/100,000
			Incidence of acute hepatitis D infection	Decrease from 3.2/100,000 to 0.04/100,000 [123]
North America	Alaska, USA	From 1984–1988 to 1999	Incidence of HCC among children	Decrease from 3/100,000 to 0/100,000 [124]
South America	Amazon Region, Colombia	Before 1992 and after 1999	Prevalence of HBV infection among children 5–9 years old	Decrease from 32% to 9%
			Children 10–14 years old	Decrease from 66% to 25% [125]
Australia and Oceania	Australia	1981–2000 to 2000–2018	HBsAg seroprevalence among Aboriginal people	Decrease from 10.8% to 3.5% [126]

HBsAg, hepatitis B surface antigen; HBV, hepatitis B virus; HCC, hepatocellular carcinoma.

**Table 7 vaccines-12-00439-t007:** Barriers to adopting and implementing hepatitis B birth dose vaccine.

Barriers to Adopting Universal Hepatitis B Birth Dose Vaccination Programs
Lack of financial support
Lack of data on prevalence of HBV infection in community to guide health policy makers
Insufficient cold chain storage
High proportion of home births
Lack of trained health care workers
Lack of political will
**Barriers in Established Hepatitis B Birth Dose Vaccination Programs**
Lack of funding or out-of-pocket payment requirements
Poor monitoring and evaluation systems
Lack of integration with the maternal and child health package
Lack of awareness about HBV infection and hepatitis B birth dose vaccination among pregnant women
Geographical inaccessibility of immunization clinics
Birth doses administered on discharge only
Inaccessibility due to allotted vaccination days
Unreliable vaccine supplies
High proportion of home births
Lack of outreach services
Mistrust of health care workers handling newborns

HBV, hepatitis B virus.

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
