# Peer review of "Hepatitis B Vaccine: Four Decades on"

_vaccines, 2024, doi:10.3390/vaccines12040439_

Round 1
Reviewer 1 Report
Comments and Suggestions for Authors
Thank you for the opportunity to review manuscript ID: vaccines-2919375.
This manuscript provides a comprehensive overview of the history and current status both of vaccines and vaccination against hepatitis B. This review paper highlights important achievements and efforts in hepatitis B vaccine development, and with questions that still await resolution. The paper is written in a clear and informative manner, citing numerous data from the literature.
In presenting key information, the authors used the most relevant sources/literature, including numerous systematic reviews and meta-analyses.
Minor comments:
- Please, check how the labeled data is presented in Figure 2.
Namely, for the United States of America, for Serbia, Albania, Algeria, etc., there is data on the mentioned reference 117. The authors stated that the WHO website was accessed on `Accessed February 9`, but the data for coverage in the period 2020-2022 can be seen on this site, and can be seen for a longer previous period and for a larger number of countries.
In addition, for example, for Brazil, the coverage data could hardly be labeled with the level in Figure 2. Please check.
- List and discuss the problems faced by countries that have not reached satisfactory hepatitis B vaccination coverage.
Author Response
Thank you for the opportunity to review manuscript ID: vaccines-2919375.
This manuscript provides a comprehensive overview of the history and current status both of vaccines and vaccination against hepatitis B. This review paper highlights important achievements and efforts in hepatitis B vaccine development, and with questions that still await resolution. The paper is written in a clear and informative manner, citing numerous data from the literature.
In presenting key information, the authors used the most relevant sources/literature, including numerous systematic reviews and meta-analyses.
Response: Thank you for the positive feedback.
Minor comments:
- Please, check how the labeled data is presented in Figure 2.
Namely, for the United States of America, for Serbia, Albania, Algeria, etc., there is data on the mentioned reference 117. The authors stated that the WHO website was accessed on `Accessed February 9`, but the data for coverage in the period 2020-2022 can be seen on this site and can be seen for a longer previous period and for a larger number of countries.
In addition, for example, for Brazil, the coverage data could hardly be labeled with the level in Figure 2. Please check.
Response: Thank you for pointing this out. The figure has been updated to reflect only most recent data for BD coverage, and it reflects the data for year 2022. The corresponding reference has been updated.
- List and discuss the problems faced by countries that have not reached satisfactory hepatitis B vaccination coverage.
Response: We have created a new table (Table 7) that lists the main barriers to adopting and implementing hepatitis B birth dose vaccine. We have also added the following statement in section 13 “Challenges in hepatitis B elimination and future steps”:
“The region with the lowest BD coverage is Sub-Saharan Africa, where the estimated HBsAg prevalence ranges from 6.3-10%.132,133 Barriers to providing BD vaccination occur at all levels: policy, facility and logistics, and community. (Table 7)”
Reviewer 2 Report
Comments and Suggestions for Authors
This paper comprehensively reviews the global impact, developments, and challenges associated with the Hepatitis B vaccine over the past four decades. Its findings are crucial for understanding the virus's transmission, history, development, and available types, including mono- and triple-antigen vaccines, and their safety profiles. The paper delves into vaccination schedules, target populations, efficacy, seroprotection rates, and non-responder management. It underscores vaccination programs' clinical and economic impact worldwide, demonstrating substantial reductions in HBV infection rates, morbidity, and mortality. Moreover, it tackles the hurdles in achieving global hepatitis B elimination, such as improved birth-dose coverage and managing hepatitis B vaccine escape mutants. The document concludes with recommendations for future strategies to enhance vaccine coverage and effectiveness.
Drawbacks:
1. The paper mentions the potential threat of vaccine-induced escape mutants, which could undermine the vaccine's effectiveness and jeopardize hepatitis B elimination efforts.
2. While the paper discusses existing vaccines and recent advancements, it could benefit from a more detailed exploration of ongoing research into new vaccine technologies and formulations that might address current limitations, such as improved efficacy in immunocompromised individuals or those with chronic kidney disease.
Recommendations:
1. Discuss the strengthening efforts to increase global birth dose coverage through improved healthcare infrastructure, awareness campaigns, and policies that facilitate access to vaccination in low- and middle-income countries.
2. Discuss research into novel vaccine technologies and adjuvants to enhance immunogenicity, particularly in hard-to-protect populations. This includes exploring new delivery mechanisms that could simplify vaccination schedules and improve compliance.
3. Discuss implementing more robust systems for collecting and analyzing vaccination and infection rate data. This will help identify areas for improvement in vaccination strategies and target interventions more effectively.
4. Highlight the novelty of the study.
Author Response
This paper comprehensively reviews the global impact, developments, and challenges associated with the Hepatitis B vaccine over the past four decades. Its findings are crucial for understanding the virus's transmission, history, development, and available types, including mono- and triple-antigen vaccines, and their safety profiles. The paper delves into vaccination schedules, target populations, efficacy, seroprotection rates, and non-responder management. It underscores vaccination programs' clinical and economic impact worldwide, demonstrating substantial reductions in HBV infection rates, morbidity, and mortality. Moreover, it tackles the hurdles in achieving global hepatitis B elimination, such as improved birth-dose coverage and managing hepatitis B vaccine escape mutants. The document concludes with recommendations for future strategies to enhance vaccine coverage and effectiveness.
Response: Thank you very much for this positive feedback.
Drawbacks:
- The paper mentions the potential threat of vaccine-induced escape mutants, which could undermine the vaccine's effectiveness and jeopardize hepatitis B elimination efforts.
Response: Despite the concern of vaccine escape mutants in reducing the effectiveness of hepatitis B vaccine, the preponderance of evidence over the last 40 years supports the vaccine’s effectiveness in preventing HBV infection and breakthrough infections caused by vaccine-escape mutants are uncommon and, therefore should not jeopardize hepatitis B elimination efforts.
- While the paper discusses existing vaccines and recent advancements, it could benefit from a more detailed exploration of ongoing research into new vaccine technologies and formulations that might address current limitations, such as improved efficacy in immunocompromised individuals or those with chronic kidney disease.
Recommendations:
- Discuss the strengthening efforts to increase global birth dose coverage through improved healthcare infrastructure, awareness campaigns, and policies that facilitate access to vaccination in low- and middle-income countries.
Response: Thank you for this comment, this section (section 13 “Challenges in hepatitis B elimination and future steps”) has been updated.
“MTCT is the most common route of transmission worldwide. Thus, disrupting MTCT is crucial for elimination of HBV. Modeling studies have reported that prenatal care for HBsAg-positive mothers and their children remains suboptimal in 2022. Only 3% of pregnant women with high viral load (>200,000 IU/mL) receive antiviral treatment during pregnancy, only 46% infants receive timely BD vaccination (Figure 2), and 14% of infants receive HBIG.132 In addition to reducing MTCT, greater efforts are needed to increased BD and “catch-up” vaccination particularly in high prevalent areas. The region with the lowest BD coverage is Sub-Saharan Africa, where the estimated HBsAg prevalence ranges from 6.3-10%.132,133 Barriers to providing BD vaccination occur at all levels: policy, facility and logistics, and community. (Table 7) Logistical constraints are mostly associated with births occurring in remote areas and outside health facilities.133 Strategies to overcome these challenges include improving healthcare infrastructure, promotion of awareness campaigns, and policies that facilitate access to vaccination such as maintaining a consistent and reliable vaccine supply, proper vaccine storage under controlled-temperature chain protocol when cold chain use is impossible, and use of prefilled auto-disposable devices to facilitate vaccine administration.137 In tandem with these solutions there is a critical need for more robust systems for collecting and analyzing vaccination and infection rate data in order to help identify areas for improving vaccination strategies and targeting interventions more effectively.”
- Discuss research into novel vaccine technologies and adjuvants to enhance immunogenicity, particularly in hard-to-protect populations. This includes exploring new delivery mechanisms that could simplify vaccination schedules and improve compliance.
Response: Thank you for this comment. We have included a new section 4 on novel vaccine approaches.
”Novel Vaccine Approaches
While current vaccines are effective, the requirement for two or three doses remains a barrier to delivery and more widespread use. Current research is aimed at developing vaccines that are more immunogenic using novel adjuvants thereby increasing seroprotection rates particularly among difficult to vaccinate populations. A marked antibody response has been observed with the ferritin nanoparticle-preS1 vaccine, which holds the potential for use both as a prophylactic and a therapeutic vaccine.25 A plasmid DNA fusion vaccine encoding mouse DEC-205 single-chain fragment variable (mDEC-205-scFv), to direct antigens to dendritic cells (DCs) through the DC-specific surface molecule DEC-205, was linked with the HBsAg.26 This vaccine was shown to induce a robust antiviral T cell and antibody immunity against HBsAg in HBV transgenic mice. Another approach is to use HBsAg virus-like particles as biotemplates to synthesize silica-adjuvanted VLP@Silica nanovaccines, that can induce both a humoral and cellular immune response.27 The success of COVID-19 vaccines has spurred further research into mRNA vaccines for hepatitis B. While initially pursued as a promising therapeutic vaccine, mRNA vaccines can induce more robust immune response and may be used for prophylaxis.28 Strategies are also focused on utilizing novel delivery systems including adenoviral and yeast vectors to improve vaccine effectiveness. Adenoviral vectors have been noted for eliciting a strong antibody response, which is particularly promising for individuals with low seroconversion rates. Moreover, adenovirus vector-based vaccines might offer advantages in terms of manufacturing ease and reduced storage costs compared to recombinant vaccines, potentially enhancing their adoption in low- and middle-income countries.29 Another novel approach to antigen delivery is to coat soluble microneedle arrays with mannose-modified poly lactic-co-glycolic acid nanoparticles. Proof-of-principle of this approach was demonstrated in a mouse model.30
Alternate routes of administration are being explored including intranasal and intradermal routes. An intranasal vaccine containing both HBsAg and hepatitis B core antigen (HBcAg) mixed with a viscosity enhancer, carboxyl vinyl polymer (CVP-NASVAC) was shown to induce anti-HBs and anti-HBc production and led to a substantial increase in HBs- and HBc-specific cytotoxic T-lymphocytes responses after CVP-NASVAC administration in prior hepatitis B recombinant vaccine non-responders. Another approach is a microneedle patch, a promising vaccine delivery method that does not require a cold chain system and is convenient for use in areas where healthcare resources are scarce; it has been shown to elicit a stronger immune response than traditional intramuscular administration.31,32”
- Discuss implementing more robust systems for collecting and analyzing vaccination and infection rate data. This will help identify areas for improvement in vaccination strategies and target interventions more effectively.
Response: Thank you very much for bringing up this relevant topic. An in-depth review of systems for collecting and analyzing vaccination and infection rate data, are beyond the scope of this review. We agree with the reviewer that these data are indispensable for policy makers to improve vaccine uptake. We have added the following sentence in section 13 “Challenges in hepatitis B elimination and future steps”:
“In tandem with these solutions there is a critical need for more robust systems for collecting and analyzing vaccination and infection rate data in order to help identify areas for improving vaccination strategies and targeting interventions more effectively.”
- Highlight the novelty of the study.
Response: This scoping review which aims to provide an update on developments, and challenges associated with the hepatitis B vaccine over the past four decades is intended for a broad audience. We have tried to highlight novel developments where appropriate.
Reviewer 3 Report
Comments and Suggestions for Authors
The review by Mironova and Ghani reports a review of HBV vaccine distribution, monitoring, efficacy and cost -effectiveness worldwide.
In page 2, 2d paragraph, it should be made clear that horizontal HBV transmission in sub-Saharan Africa and elsewhere is by contacts between toddlers and young children being infected at pre-school or school before age 10. This mode far outnumbers family contacts and transfusion accounting for less than 10% altogether. This early vertical or horizontal transmissions explain the high prevalence of chronic infections.
The authors should mention the efforts of Maupas P et al. in the late 70s developing a plasma derived HBV vaccine ( Immunisation against hepatitis B in man. Lancet. 1976;1:1367-70.
In section 5, it should be clearly indicated that combined immunisations are typically scheduled at 6 weeks of life and the following months, a delay that might impact the efficacy of vaccine. It is the schedule in sub-Saharan Africa where prevalence is very high while vaccination at birth is universally practiced in Asia.
In section 7, authors should report on multiple articles, mostly from China and Taiwan, examining the impact of HBV vaccination in adults by screening for anti-HBc and finding substantial prevalence of marker of post-vaccine contacts with HBV determining anti-HBc response. Low or undetectable anti-HBs, poorly managed vaccination, or lack of evidence of vaccination were blamed but calls for boosting vaccine injection at adolescence (before start of sexual contacts) were advocated.
Wang Z, et al. Prevalence of hepatitis B surface antigen (HBsAg) in a blood donor population born prior to and after implementation of universal vaccination in Shenzhen, China. BMC Inf Dis 2016; 16: 498-507.
Tang X, et al. Incidence of HBV infection in young Chinese blood donors born after mandatory implementation of neonatal hepatitis B vaccination nationwide. J Viral Hepat. 2018; 25: 1-9.
Anti-HBc-nonreactive occult hepatitis B infections with HBV genotypes B and C in vaccinated immunocompetent adults.J Viral Hepat. 2022;29:958-967.
In this section, the authors should search the literature to determine the efficacy of vaccination at 6 weeks as in Africa and compare it to birth vaccination as in Asia in terms of anti-HBs response and long-term protection.
In section 10, it would be useful to provide a table showing the amino acid substitutions of the main vaccine escape mutants and their prevalence.
In section 11.2, it should be indicated that in many subs-Saharan African countries, cost of HBV vaccine is often supported by families and a limitation to HBV vaccine expansion.
Comments on the Quality of English LanguageGenerally good, only need typos corrections.
Author Response
The review by Mironova and Ghany reports a review of HBV vaccine distribution, monitoring, efficacy and cost -effectiveness worldwide.
In page 2, 2d paragraph, it should be made clear that horizontal HBV transmission in sub-Saharan Africa and elsewhere is by contacts between toddlers and young children being infected at pre-school or school before age 10. This mode far outnumbers family contacts and transfusion accounting for less than 10% altogether. This early vertical or horizontal transmissions explain the high prevalence of chronic infections.
Response: Thank you very much for this suggestion. We have updated the section 2 “Hepatitis B-Virus and Hepatitis B Disease”.
“In regions of high HBV prevalence there are geographical differences in mode of HBV transmission. The predominant mode of transmission in Asia is vertical due to a high prevalence of hepatitis B e antigen (HBeAg) positivity among women of childbearing potential whereas in sub-Saharan Africa the primary mode of transmission is horizontal between toddlers and children with most being infected by school age.”
The authors should mention the efforts of Maupas P et al. in the late 70s developing a plasma derived HBV vaccine (Maupas P, Goudeau A, Coursaget P, Drucker J, Bagros P. Immunisation against hepatitis B in man. Lancet. 1976;1:1367-70.
Response: The requested reference was added.
“Independently, Purcell and Gerin at National Institute of Allergy and Infectious Diseases, Hilleman at Merck, and Maupas at the Institute de Virologie de Tours isolated and purified HBsAg from the plasma of HBsAg positive persons and tested immunogenicity and efficacy in chimpanzees.11-13”
In section 5, it should be clearly indicated that combined immunisations are typically scheduled at 6 weeks of life and the following months, a delay that might impact the efficacy of vaccine. It is the schedule in sub-Saharan Africa where prevalence is very high while vaccination at birth is universally practiced in Asia.
Response: Thank you for the suggestion. We have revised the section 6, “Immunization schedules”.
“In Africa only 14 out of 47 countries have introduced BD vaccination into their immunization calendars,45 with a majority of countries starting immunization at 6 weeks. Although vertical transmission is a less common mode of transmission in Africa, delay in vaccine administration may contribute to incident infection and timely administration of BD vaccination is preferred.”
In section 7, authors should report on multiple articles, mostly from China and Taiwan, examining the impact of HBV vaccination in adults by screening for anti-HBc and finding substantial prevalence of marker of post-vaccine contacts with HBV determining anti-HBc response. Low or undetectable anti-HBs, poorly managed vaccination, or lack of evidence of vaccination were blamed but calls for boosting vaccine injection at adolescence (before start of sexual contacts) were advocated.
Wang Z, et al. Prevalence of hepatitis B surface antigen (HBsAg) in a blood donor population born prior to and after implementation of universal vaccination in Shenzhen, China. BMC Inf Dis 2016; 16: 498-507.
Tang X, et al. Incidence of HBV infection in young Chinese blood donors born after mandatory implementation of neonatal hepatitis B vaccination nationwide. J Viral Hepat. 2018; 25: 1-9.
Deng X, et al. Anti-HBc-nonreactive occult hepatitis B infections with HBV genotypes B and C in vaccinated immunocompetent adults.J Viral Hepat. 2022;29:958-967.
Response: Thank you for this issue. It was added to the review, section 8 “Durability of seroprotection and need for booster dose”, second paragraph.
“…several studies on blood donors in China suggesting that young individuals who have been vaccinated remain susceptible to occult HBV infections, as indicated by presence of isolated anti-HBc, anti-HBc and anti-HBs and even anti-HBs alone together with intermittent low-level detection of HBV DNA suggesting vaccine failure in some cases. This has been attributed to waning immunity with low or undetectable anti-HBs, failure to complete the vaccination schedule, or lack of evidence of vaccination leading some to recommend a booster vaccine at adolescence (before start of sexual contacts). The adoption of such a strategy remains controversial and is not currently recommended 68-70 Moreover, a study from Thailand compared long-term seroprotection during a 20-year follow-up after vaccination in infancy, where half of the subjects received a booster. The vaccine provided long-term seroprotection regardless of whether a booster dose was administered, although there was evidence of HBV exposure in the cohort. Breakthrough infections in the first decade usually occurred due to household exposure, while those in the second decade occurred with beginning of adolescent behaviors.71
In this section, the authors should search the literature to determine the efficacy of vaccination at 6 weeks as in Africa and compare it to birth vaccination as in Asia in terms of anti-HBs response and long-term protection.
Response: Thank you very much for this suggestion. This information has been added to section 8, “Durability of seroprotection and need for booster dose”:
“A meta-analysis of studies conducted to assess seroprotection rates based on the timing of vaccination in children, reported the pooled seroprotection rate in children under 5 years old in Africa with four doses (i.e., a BD and three doses) was significantly higher than three doses, highlighting the importance of BD.65 Another study from Pakistan compared seroprotection rates 8 weeks after completion of a vaccination series in infants who received BD and three doses of combination vaccine and those, who received combination vaccines only. There was a dramatic difference observed between the BD and non-BD groups, 95.9% vs. 58.8%, respectively.66 In a study from the Philippines, seroprotection rates one month after vaccination series with DTPw-HBV/Hib was higher in infants who also received a BD vaccine, 94.3% compared to 87.6% in infants who received the combination vaccine only.67 Taken together, these data provide support for the importance of incorporating of BD into national immunization calendars worldwide.”
In section 10, it would be useful to provide a table showing the amino acid substitutions of the main vaccine escape mutants and their prevalence.
Response: We have provided a new table, Table 5, listing the amino acid substitutions of the main vaccine escape mutants. It is difficult to determine the prevalence of each of these mutations due to a paucity of good data. We have highlighted the most common mutation the G145R.
In section 11.2, it should be indicated that in many subs-Saharan African countries, cost of HBV vaccine is often supported by families and a limitation to HBV vaccine expansion.
Response: We have included this in a new Table 7 listing barriers to birth dose vaccination.
Reviewer 4 Report
Comments and Suggestions for Authors
This is an excellent scientific and historical overview of hepatitis B vaccines. I only have minor comments and suggestions:
Line 30: The data are strongest for prevention of childhood HCCC, but there is no reason why HBV vaccination wouldn’t prevent HBV-associated HCC in anyone with effective vaccine-induced seroprotection.
Line 46: Are there any human cells that support HBV replication outside hepatocytes?
Line 51: “…targeted by neutralizing antibodies anti-HBs.” Looks odd. Maybe (anti-HBs)?
Line 59: “…approximately 30% of persons present with jaundice” – infants and children under age 5 with initial infection have a much lower incidence of jaundice.
Line 128: Screening for HBV is separate from recommendations on HBV vaccination – one can have a vaccination program without a screening program, so can these two recommendations be separated here?
Table 1: The actual ACIP recommendations state "Persons with diabetes at the discretion of the treating clinician". This specific wording is subtle but may have implications for electronic health decision support. This Table is only a partial list of risk factors - can that be stated in case readers want to see the entire list of risk factors? It needs to be left-justified. Currently is difficult to read.
Line 157: BD was not clearly defined prior to use of this abbreviation.
Line 169: “There was no difference between those who received [a] second dose within [an] interval [of] 4-8 weeks and after 8 weeks”
Line 169: The approved two-dose HebB-CpG is not labeled for a third dose to provide long-term protection.
Line 196: Seroprotection rates for adults ages 18-44 were 99.2% with PreHevbrio and Heplisav-B reported rates of 100% for ages 18-29, 98.9% for ages 30-39, and 97.2% for ages 40-49 per their package inserts. These higher rates are important because we preferentially use these two vaccines in high risk settings such as syringe access programs.
Line 224: This amnesiac response as well as cellular immunity contributes to the lack of progression to chronic active infection in those with a history of ever mounting an effective humoral response to HBV vaccination, but they can develop anti-HBc. Are these people at risk for reactivation if immunosuppressed in the future? Would the presence of measurable anti-HBs reduce this risk? Screening programs identify many adults with undetectable anti-HBs levels and primary care providers feel like not boosting them is counter-intuitive.
Lines 334, 341: Is this rate of vaccine escape mutants representative of the rate of vaccine failure? It is hard to follow how many people who received HBV vaccines but still developed chronic infection did so because of mutations that reduced the effectiveness of the vaccine.
Line 355: Risk-based vaccination was a failure, which is why ACIP now recommends vaccination of all adults <age 60 as a “catch up” effort, without regard to risk factors. Others countries would probably benefit from a similar strategy.
Table 5: The components of this table are not aligned and it is hard to read
Line 420: remains
Line 424: is
Line 424: “…occurring in communities out of traditional healthcare systems.” Traditional can mean non-formal healthcare settings. It might be clearer as “…occurring outside health facilities” (see HV Doctor, 2018, BMC Public Health)
Author Response
This is an excellent scientific and historical overview of hepatitis B vaccines. I only have minor comments and suggestions:
Line 30: The data are strongest for prevention of childhood HCCC, but there is no reason why HBV vaccination wouldn’t prevent HBV-associated HCC in anyone with effective vaccine-induced seroprotection.
Response: In principle we agree with the reviewer. However, the earliest vaccine program did not begin until 1984 therefore there are no data on persons beyond the age of 40 years. We would prefer not to make statements that are non-evidenced based. It is possible that if vaccine-induced immunity were to wane and persons contracted HBV, such persons may be at risk for HCC later in life. We have edited the sentence in section 12 “Clinical and economic impact of hepatitis B vaccination”, “Furthermore, the incidence rate of HCC has decreased by more than 80% in the population aged 5-29 years and likely would be effective in older persons as long as there is evidence of seroprotection.”
Line 46: Are there any human cells that support HBV replication outside hepatocytes?
Response: HBV infection can only be established in human hepatocytes due to the requirement for sodium taurocholate co-transporting polypeptide (NTCP) and other human hepatocyte restriction factor(s). It is possible infect non-human hepatocytes by artificially expressing NTCP but not on other cell types.
Line 51: “…targeted by neutralizing antibodies anti-HBs.” Looks odd. Maybe (anti-HBs)?
Response: Thank you for pointing this out. It was corrected:
“…targeted by neutralizing antibodies (anti-HBs).”
Line 59: “…approximately 30% of persons present with jaundice” – infants and children under age 5 with initial infection have a much lower incidence of jaundice.
Response: We have edited the sentence:
“The initial infection is typically asymptomatic and anicteric particularly when infection is acquired vertically. Approximately 30% of adults may present with jaundice; acute liver failure is uncommon but occurs in both adults and children.”
Line 128: Screening for HBV is separate from recommendations on HBV vaccination – one can have a vaccination program without a screening program, so can these two recommendations be separated here?
Response: We agree with the reviewer on this point. We have separated the two recommendations.
“In 2023, the Centers for Disease Control and Prevention (CDC) updated their recommendations advising hepatitis B vaccine to all adults through age 59 years who have not been vaccinated or whose vaccination status is unknown and adults age 60 years or older with risk factors for hepatitis B infection. Additionally, the CDC recommended that all adults should be screened for HBV at least once in their lifetime.”
Table 1: The actual ACIP recommendations state "Persons with diabetes at the discretion of the treating clinician". This specific wording is subtle but may have implications for electronic health decision support. This Table is only a partial list of risk factors - can that be stated in case readers want to see the entire list of risk factors? It needs to be left-justified. Currently is difficult to read.
Response: Thank you for the comment. Table 1 has been updated to include all the recommendations on who should receive hepatitis B vaccine in the United States. Formatting issues have been corrected.
Line 157: BD was not clearly defined prior to use of this abbreviation.
Response: The acronym for birth dose (BD) was provided in the introduction. Second paragraph, last line, “…birth dose (BD) vaccine remains suboptimal.”
Line 169: “There was no difference between those who received [a] second dose within [an] interval [of] 4-8 weeks and after 8 weeks”
Response: Grammar was corrected.
Line 169: The approved two-dose HebB-CpG is not labeled for a third dose to provide long-term protection.
Response. We agree that HebB-CpG is not licensed for a third dose to provide long-term protection. The paragraph was discussing the recombinant vaccine for infants, not HepB-CpG. We have updated the sentence to avoid confusion, “The third dose of recombinant vaccine confers…”
Line 196: Seroprotection rates for adults ages 18-44 were 99.2% with PreHevbrio and Heplisav-B reported rates of 100% for ages 18-29, 98.9% for ages 30-39, and 97.2% for ages 40-49 per their package inserts. These higher rates are important because we preferentially use these two vaccines in high-risk settings such as syringe access programs.
Response: We have updated the section 7, “Efficacy and seroprotection rates”:
“Seroprotection rates in adults are higher with the newer vaccines compared to Engerix, PreHevbrio (91.4 versus 76.5%, respectively) and Heplisav-B (95.4% versus 81.3%, respectively) and particularly in adults older than 50 years, making them an attractive choice for more difficult to vaccinate persons.”
Line 224: This amnesiac response as well as cellular immunity contributes to the lack of progression to chronic active infection in those with a history of ever mounting an effective humoral response to HBV vaccination, but they can develop anti-HBc. Are these people at risk for reactivation if immunosuppressed in the future? Would the presence of measurable anti-HBs reduce this risk? Screening programs identify many adults with undetectable anti-HBs levels and primary care providers feel like not boosting them is counter-intuitive.
Response: Persons with isolated anti-HBc are at risk for HBV reactivation if receiving immunosuppression or chemotherapy classified as high risk for reactivation e.g. anti-CD20 therapy. A level of anti-HBs >100 IU/mL has been shown to be associated with a reduced risk for reactivation. J. Med. Virol. 2016;88:1010–1017; Cancer. 2010;116:4769–4776. Currently, guidelines do not recommend vaccination to boost anti-HBs levels to >100 IU/mL as a strategy to prevent HBV reactivation. This is partly due to lack of high-quality data and the general high effectiveness of prophylactic antiviral therapy. Additionally, as shown in a small study among patients without serological markers of HBV scheduled to receive rituximab therapy, no patient developed anti-HBs when hepatitis B vaccine was administered a median of 2 weeks prior to initiating rituximab. Viruses. 2022 Aug; 14(8): 1780. This suggest that a longer lead time is likely required to induce anti-HBs. Whether the newer generation vaccines could improve sero-protection rates in this population remains to be determined. Thus, at present we cannot make a recommendation to boost persons with isolated anti-HBc.
We have added the following sentences that address the issue in section 8, “Durability of seroprotection and need for booster dose”:
“Persons with isolated anti-HBc are at risk for HBV reactivation if receiving immunosuppressive therapy. American Association for the Study of Liver Diseases Hepatitis B guidance does not currently recommend vaccination to boost anti-HBs levels to >100 IU/mL as a strategy to prevent HBV reactivation among persons scheduled to receive immunosuppressive therapy. The European Association for the Study of Liver Diseases hepatitis B guidelines does recommend that patients without serological evidence on Hepatitis B and who are candidates for immunosuppression should be vaccinated.73”
Lines 334, 341: Is this rate of vaccine escape mutants representative of the rate of vaccine failure? It is hard to follow how many people who received HBV vaccines but still developed chronic infection did so because of mutations that reduced the effectiveness of the vaccine.
Response: We have added a sentence to clarify the issue in section 11, “Hepatitis B vaccine escape mutants”:
“While vaccine escape mutations may account for some vaccine failures, the prevalence of vaccine escape mutations are lower than the rate of vaccine failures suggesting that these mutations cannot account for all immunoprophylactic failures in infants.”
Line 355: Risk-based vaccination was a failure, which is why ACIP now recommends vaccination of all adults <age 60 as a “catch up” effort, without regard to risk factors. Others countries would probably benefit from a similar strategy.
Response: We agree with the reviewer. We have updated the section 12, “Clinical and economic impact of hepatitis B vaccination”:
“Therefore, in addition to universal vaccination of infants beginning at birth, the Advisory Committee on Immunization Practices (ACIP) recommends “catch-up” vaccination among all adults aged 19–59 years and adults > 60 years with or without risk factors for hepatitis B who seek protection.”
Table 5: The components of this table are not aligned and it is hard to read
Response: The Table has been reformatted.
Line 420: remains
Response: This has been corrected.
Line 424: is
Response: This has been corrected.
Line 424: “…occurring in communities out of traditional healthcare systems.” Traditional can mean non-formal healthcare settings. It might be clearer as “…occurring outside health facilities” (see HV Doctor, 2018, BMC Public Health)
Response: Thank you, this has been corrected to “occurring outside health facilities”.
Round 2
Reviewer 3 Report
Comments and Suggestions for Authors
The authors have adequately incorporated this reviewer's comments significantly improving their review.
Comments on the Quality of English LanguageAdequate